# Identifying the Metabolic Signatures of PPARD-Overexpressing Gastric Tumors

**DOI:** 10.3390/ijms23031645

**Published:** 2022-01-31

**Authors:** Shivanand Pudakalakatti, Mark Titus, José S. Enriquez, Sumankalai Ramachandran, Niki M. Zacharias, Imad Shureiqi, Yi Liu, James C. Yao, Xiangsheng Zuo, Pratip K. Bhattacharya

**Affiliations:** 1Department of Cancer Systems Imaging, The University of Texas MD Anderson Cancer Center, Houston, TX 77030, USA; spudakalakatti@mdanderson.org (S.P.); Jose.EnriquezOrtiz@uth.tmc.edu (J.S.E.); 2Department of Genitourinary Medical Oncology, The University of Texas MD Anderson Cancer Center, Houston, TX 77030, USA; MTitus1@mdanderson.org (M.T.); SRamachandran@mdanderson.org (S.R.); 3MD Anderson Cancer Center, UTHealth Graduate School of Biomedical Sciences, Houston, TX 77030, USA; NMZacharias@mdanderson.org; 4Department of Urology, The University of Texas MD Anderson Cancer Center, Houston, TX 77030, USA; 5Department of Gastrointestinal Medical Oncology, The University of Texas MD Anderson Cancer Center, Houston, TX 77030, USA; ishureiq@med.umich.edu (I.S.); YLiu25@mdanderson.org (Y.L.); jyao@mdanderson.org (J.C.Y.); xzuo@mdanderson.org (X.Z.)

**Keywords:** gastric cancer, PPARD, NMR spectroscopy, LC-MS, hyperpolarized [1-^13^C] pyruvate MR spectroscopy, metabolomics, metabolites, metabolic imaging

## Abstract

Peroxisome proliferator-activated receptor delta (PPARD) is a nuclear receptor known to play an essential role in regulation of cell metabolism, cell proliferation, inflammation, and tumorigenesis in normal and cancer cells. Recently, we found that a newly generated villin-PPARD mouse model, in which PPARD is overexpressed in villin-positive gastric progenitor cells, demonstrated spontaneous development of large, invasive gastric tumors as the mice aged. However, the role of PPARD in regulation of downstream metabolism in normal gastric and tumor cells is elusive. The aim of the present study was to find PPARD-regulated downstream metabolic changes and to determine the potential significance of those changes to gastric tumorigenesis in mice. Hyperpolarized [1-^13^C] pyruvate magnetic resonance spectroscopy, nuclear magnetic resonance spectroscopy, and liquid chromatography-mass spectrometry were employed for metabolic profiling to determine the PPARD-regulated metabolite changes in PPARD mice at different ages during the development of gastric cancer, and the changes were compared to corresponding wild-type mice. Nuclear magnetic resonance spectroscopy-based metabolomic screening results showed higher levels of inosine monophosphate (*p* = 0.0054), uracil (*p* = 0.0205), phenylalanine (*p* = 0.017), glycine (*p* = 0.014), and isocitrate (*p* = 0.029) and lower levels of inosine (*p* = 0.0188) in 55-week-old PPARD mice than in 55-week-old wild-type mice. As the PPARD mice aged from 10 weeks to 35 weeks and 55 weeks, we observed significant changes in levels of the metabolites inosine monophosphate (*p* = 0.0054), adenosine monophosphate (*p* = 0.009), UDP-glucose (*p* = 0.0006), and oxypurinol (*p* = 0.039). Hyperpolarized [1-^13^C] pyruvate magnetic resonance spectroscopy performed to measure lactate flux in live 10-week-old PPARD mice with no gastric tumors and 35-week-old PPARD mice with gastric tumors did not reveal a significant difference in the ratio of lactate to total pyruvate plus lactate, indicating that this PPARD-induced spontaneous gastric tumor development does not require glycolysis as the main source of fuel for tumorigenesis. Liquid chromatography-mass spectrometry-based measurement of fatty acid levels showed lower linoleic acid, palmitic acid, oleic acid, and steric acid levels in 55-week-old PPARD mice than in 10-week-old PPARD mice, supporting fatty acid oxidation as a bioenergy source for PPARD-expressing gastric tumors.

## 1. Introduction

Gastric cancer (GC) is the fifth most common malignancy and third most lethal cancer worldwide, with a 5-year survival rate of 5–10% for GC in advanced stages [1,2,3]. The 5-year survival rate for GC has not markedly improved despite decades of intensive research. Effective strategies for GC prevention and treatment, as well as understanding the drivers of this often fatal disease, are therefore greatly needed. Identification of critical events in GC initiation and progression could provide crucial knowledge for developing mechanism-based intervention strategies to improve GC outcomes.

Peroxisome proliferator-activated receptor delta (PPARD) is a ligand-dependent nuclear transcription factor that regulates a multiplicity of pathophysiological processes associated with glucose and lipid metabolism, cell proliferation, differentiation, and inflammation [4]. Several synthetic ligands, including GW501516 and GW0742, selectively bind and activate PPARD, changing the body’s fuel preference from glucose to lipids and increasing the endurance of muscle cells. Thus, researchers have suggested use of these ligands for treatment of obesity and dyslipidemia and for weight loss [5]. Surprisingly, GW501516 was found to have a tumor-promoting function in preclinical animal models [6,7,8], and fatty acids in high-fat diets (e.g., arachidonic acid, linoleic acid (LA), and palmitic acid (PA)), serving as natural activating ligands of PPARD [9,10], transformed APC-mutant progenitor cells and promoted colorectal tumorigenesis [6]. All of these data collectively raise concerns about the safety of using PPARD agonists clinically. Specifically, researchers have questioned the direction of PPARD’s effects on tumorigenesis, primarily because of studies of germ-line PPARD knockout in Apc^min^ mice showing conflicting results regarding the role of PPARD in intestinal tumorigenesis [11,12]. Nevertheless, PPARD is upregulated in many cancers, including breast [8,13], colon [14,15], lung [16], and head and neck [17] cancers. Several oncogenic signaling pathways positively regulate PPARD, including K-Ras [18], Wnt [19], and Src [20]. Disruption of PPARD expression in cancer cells suppresses the development and metastasis of mammary, pancreatic, and colon cancer and melanoma [15,21,22,23].

Although increasing data suggest that PPARD contributes to the development of GC, definitive evidence of this is still lacking, partially because of the limitations of current *in vivo* experimental modeling, especially in relation to the most common GC type, adenocarcinoma. Recently, we found that genetically engineered mice with targeted gut overexpression of PPARD driven by the villin promoter (villin-PPARD) experienced spontaneous development of large, invasive gastric corpus adenocarcinoma (intestinal type). Further characterization of the model in that study revealed that GC was associated with severe chronic inflammation that faithfully recapitulated the features of human GC [24]. However, the molecular mechanisms by which PPARD spontaneously induced GC development, especially in relation to PPARD-mediated metabolic changes given that PPARD is a sensor that regulates metabolism of fatty acids, remained largely unknown in that GC model. As described herein, we addressed this knowledge gap using nuclear magnetic resonance (NMR) spectroscopy, liquid chromatography-mass spectrometry (LC-MS), and hyperpolarized [1-^13^C] pyruvate magnetic resonance (HP-MR) spectroscopy to unravel the downstream metabolic profiling changes governed by PPARD in this unique villin-PPARD mouse model. Hyperpolarization allows for significant magnetic resonance signal enhancement: greater than 10,000-fold over that with Boltzmann polarization in conventional Magnetic Resonance (MR). This enhancement is preserved on the downstream metabolites of the hyperpolarized compound, enabling direct observation of metabolic flux *in vivo* in real time.

## 2. Results

Whole gastric corpus tissue samples from 10- and 55-week-old PPARD mice and their sex- and age-matched wild-type (WT) littermates were collected and subjected to *ex vivo* NMR spectroscopy and *in vivo* HP-MR spectroscopy for metabolite profiling [25,26,27,28]. We observed that gastric corpus tissues from PPARD mice at age 10 weeks (before GC developed) had levels of 23 examined metabolites (alanine, lactate, taurine, T-choline, adenosine monophosphate (AMP), imidazole, oxypurinol, nicotinurate, tyrosine, fumarate, inosine monophosphate (IMP), inosine, UDP-glucose, uracil, glucose, phenylalanine, glycine, valine, leucine, creatine, isocitrate, glutamine, and glutamate) similar to those in their WT littermates (Figure 1). In 35-week-old PPARD mice and their WT littermates, there were no significant differences in metabolite levels except for glucose (*p* = 0.01) and imidazole (*p* = 0.0011) levels (Figure 2). By week 55, we observed significantly higher IMP (*p* = 0.0054), uracil (*p* = 0.0205), phenylalanine (*p* = 0.017), glycine (*p* = 0.014), and isocitrate (*p* = 0.029) and lower inosine (*p* = 0.018) levels in PPARD mice than in their WT littermates (Figure 3). Furthermore, gastric corpus tissues from 55-week-old PPARD mice with advanced GC had significantly higher IMP (*p* = 0.011), UDP-glucose (*p* = 0.0006), and AMP (*p* = 0.009) levels than did those from 10-week-old PPARD mice (Figure 4). However, 35-week-old PPARD mice had significantly higher IMP (*p* = 0.0007), UDP-glucose (*p* = 0.0031), and oxypurinol (*p* = 0.039) levels than did 10-week-old PPARD mice. An *in vivo* hyperpolarized [1-^13^C] pyruvate MR spectroscopy study showed that PPARD mice had no significant differences in real-time pyruvate-to-lactate flux in their stomachs between the ages of 10 and 35 weeks (Figure 5A–C). The metric for dynamic pyruvate-to-lactate conversion, denoted as the normalized lactate ratio (nLac), is the ratio of lactate to lactate plus pyruvate. The nLac is determined by integrating the hyperpolarized ^13^C signal for lactate over the combined hyperpolarized ^13^C signal for lactate and pyruvate. No change in the nLac, despite measurable tumor growth from 10 to 35 weeks, was observed (Figure 5A,B,D). To the best of our knowledge, this is the first reported real-time demonstration of lack of any change in metabolic flux with the increase in tumor size. This underscores the importance of using hyperpolarized metabolic imaging to validate different metabolic mechanisms underwriting tumor biology.

To evaluate longitudinal fatty acid levels in mouse GC tumors with overexpression of PPARD, the levels of four, long-chain fatty acids were measured *ex vivo* using quantitative LC-MS. Specifically, we measured two PPARD fatty acid ligands [palmitic acid (PA) and linoleic acid (LA)] and two long-chain fatty acids [steric acid (SA) and oleic acid (OA)] in gastric corpus tissues from mice at the ages of 10 without GC development and 55 weeks with advanced GC tumors (Figure 6). In stomach tissues of 10-week-old PPARD mice, the mean PA and LA levels were 3688.5 ng/g (range: 148.9–9292.8 ng/g) and 59,135.0 ng/g (range, 4245.9–123,336.7 ng/g), respectively. However, in GC tissues of 55-week-old PPARD mice, the mean PA level decreased to 214.8 ng/g (range: 3.7–528.0 ng/g), likewise the mean LA level decreased to 4112.4 ng/g (range: 61.3–12,252.1 ng/g). Although in 55-week-old PPARD mice GC PA levels decreased 17.1-fold and LA levels decreased 14.4-fold, LA at the measured level of 4112.4 ng/g may continue to bind and activate PPARD-driven transcription in the GC tumors. The mean OA and SA levels in stomach tissue of 10-week-old PPARD mice were 15,355.1 ng/g (range: 904.4–37,808.0 ng/g) and 3358.3 ng/g (range: 2649.2–9513.0 ng/g), respectively. The OA and SA levels in GC tissue of 55-week-old PPARD mice were 16.9-fold and 12.6-fold lower, respectively, than those in stomach tissue of 10-week-old PPARD mice.

## 3. Discussion

PPARDs are ligand-activated transcription factors that regulate fatty acid metabolism. Authors have described the structure, mechanism of action, and characteristics of PPARD in the literature [29,30,31]. Moreover, researchers have studied the role of PPARD in obesity and diabetes and its action in adipose tissue, skeletal muscle, macrophages, and atherosclerosis [32]. However, the role of PPARD in the development of GC and other types of cancer is still elusive. Investigators found that PPARD expression was higher in human rectal and gastric tumors than in adjacent normal mucosa [24,33]. Furthermore, PPARD agonists have promoted the expression of vascular endothelial growth factor (VGEF) in the colorectal cancer cell lines HT29 and SW480 [34,35]. In our GC mouse model, PPARD overexpression induced this cancer. We investigated the downstream metabolic changes due to PPARD and compared them with those in WT littermates. We also measured time-dependent changes in metabolism in the PPARD mouse model using hyperpolarized [1-^13^C] pyruvate MR spectroscopy. Of the 23 metabolites analyzed, we observed significantly upregulated AMP, IMP, and UDP-glucose as GC developed in PPARD mice in an NMR spectroscopy-based metabolomic study of gastric corpus tissue samples (Figure 4). These data demonstrate metabolic rewiring and may suggest the involvement of AMPK, as it also stimulates β-oxidation of fatty acids [36,37,38]. A parallel LC-MS study revealed that PA, LA, SA, and OA levels were markedly lower in 55-week-old PPARD mice than in 10-week-old PPARD mice (Figure 6). These fatty acids provide bioenergy for GC growth via peroxisome and mitochondrial fatty acid oxidation, and the pool size for these metabolites is depleted with GC development in PPARD mice from 10 to 55 weeks of age. On the other hand, our NMR spectroscopy results showed higher AMP, IMP, and UDP-glucose levels in 55-week-old PPARD mice than in 10-week-old PPARD mice (Figure 4), which complements the LC-MS results demonstrating concomitant decreases in PA, LA, SA, and OA concentrations (Figure 6).

An important point is that we observed no changes in dynamic real-time metabolic conversion of pyruvate to lactate in hyperpolarized metabolic imaging (Figure 5A–C) despite noting measurable growth in tumor size from 10 to 35 weeks in PPARD mice (Figure 5D). Hyperpolarized [1-^13^C] pyruvate MR spectroscopy is a noninvasive metabolic imaging modality that probes carbon flux in tissues and infers the state of metabolic reprogramming in tumors. Elevated hyperpolarized pyruvate-to-lactate conversion rates (nLac) in aggressive tumors are generally attributed to enhanced glycolytic flux and lactate dehydrogenase A activity. Conventional orthodoxy holds that most cancers exhibit this enhanced aerobic glycolysis, known as the Warburg effect [39], and investigators have identified enhanced glucose uptake and use as a cancer-associated metabolic signature [40,41,42,43]. Notably, the Warburg effect is not the dominant metabolic pathway in PPARD-expressing GC, as evidenced by our real-time *in vivo* HP-MR data. Taken together, our complementary *ex vivo* NMR and LC-MS data paint a consistent picture of different, PPARD-driven metabolic rewiring in GC. We found that fatty acids were rapidly utilized to meet the energy demands of gastric tumorigenesis. This was supported by the downregulated fatty acid levels and increased amounts of energy currency (AMP and IMP). AMP and IMP also help in DNA synthesis attributed to purine metabolism, which in turn is important for proliferation of cancer cells. Our NMR spectroscopy and LC-MS results confirmed the vital role of fatty acids and purine metabolism in GC development.

NMR spectroscopy- and LC-MS-based metabolomics revealed significant changes in PPARD-regulated downstream metabolites in preclinical mouse models of GC. Unlike in many cancer systems, we found that PPARD-overexpressing GC was not primarily dependent on aerobic glycolysis for fuel. Instead, our data demonstrated that fatty acid oxidation through metabolic rewiring is the dominant bioenergy source in these tumors. PPARD and PPARD-mediated metabolite changes may be targets for developing interventions for GC chemoprevention and chemotherapy.

## 4. Materials and Methods

### 4.1. Generation of the Villin-PPARD Mouse Model and Gastric Tissue Collection

For this prospective study, PPARD mice were generated via pronuclear injection of a mouse *PPARD* expression construct under the control of a villin promoter (p12.4Kvill–*PPARD*) into fertilized FVB oocytes at the Genetically Engineered Mouse Facility at The University of Texas MD Anderson Cancer Center [24,44]. These mice and their sex-matched WT littermates (*n* = 5 per group) were longitudinally followed at 10, 35, and 55 weeks and then euthanized. Their stomachs were examined for gastric tumorigenesis, weighed, and photographed. Gastric corpus tissues were collected from PPARD and WT mice (50 mg/mouse) for further studies. The collected tissues were preserved at −80 °C until processing for NMR spectroscopy and MS data acquisition. PPARD mice at the ages of 10 and 35 weeks (*n* = 5 per group) were used for dynamic [1-^13^C] pyruvate metabolism measurements. Mice were housed in our pathogen-free facility in accordance with AAALAC International guidelines. All experiments were performed according to protocols approved by the MD Anderson Institutional Animal Care and Use Committee.

### 4.2. NMR Data Acquisition

The mouse tissues were weighed at the approximate liquid nitrogen temperature before being subjected to extraction of metabolites. After weighing, tissues were mixed with 0.5 mL of Lysing Matrix D bulk beads (cat. no. 6540434; Millipore, Bedford, MA, USA) and 3 mL of a methanol and water mixture (2:1). The mixture was vortexed for 1 min and freeze-thawed; the vortexing and freezing-thawing were repeated three times. The mixture was centrifuged at 4000 rpm for 10 min at 4 °C, and the supernatant was collected for rotary evaporation to remove methanol and water. This was followed by overnight lyophilization to remove residual water. The lyophilized sample was dissolved in 800 µL of ^2^H_2_O and centrifuged at 10,000 RPM for 5 min at room temperature to remove any debris. The 600 µL of sample was placed in an NMR tube and mixed with 40 µL of 8 mM DSS to acquire NMR data.

All NMR spectra were acquired using a Bruker AVANCE III HD NMR scanner (Bruker BioSpin MRI GmbH, Ettingen, Germany) at a temperature of 298 °K. The scanner operates at a ^1^H resonance frequency of 500 MHz and is equipped with a cryogenically cooled triple resonance (^1^H, ^13^C, and ^15^N) Prodigy broadband BBO probe with a *Z*-axis shielded gradient for increased sensitivity. One-dimensional ^1^H spectra were acquired using 1D NOESY with water suppression, 256 averages, 64 receiver gain, a 16-ppm spectral width, a 6-s relaxation delay, a 90° pulse of 12 μs, and 32k time domain points.

### 4.3. Ex Vivo Metabolite Pool Size Measurements

The raw NMR spectrometer FID files were imported into the TopSpin 3.1 software program (Bruker BioSpin MRI GmbH, Ettlingen, Germany). The spectra were manually phase-corrected to form Lorentzian peak shapes, and exponential apodization of line broadening of 0.3 Hz was applied to increase the signal-to-noise ratio. The spectra were automatically baseline-corrected (command absn) prior to peak intensity measurements [45]. Intensities were normalized to tissue weight and the reference compound 2,2-dimethyl-2-silapentane-5-sulfonate-d6 (DSS-d6) at 0.5 mM. The DSS reference peak was set to 0 ppm before integration of peaks [46]. Zero-filling of 64k was used to process the data in TopSpin 3.1 to smoothen the peaks in the spectra. The processed spectra were exported to Chenomx NMR Suite 8.1 software (Chenomx Inc., Edmonton, AB, Canada); the peaks were then identified by matching them with spectral models of metabolites contained in the Human Metabolic Database [47].

### 4.4. Magnetic Resonance Imaging and Hyperpolarized MR Data Acquisition

[1-^13^C] pyruvic acid was hyperpolarized using a mixture of 20 μL of [1-^13^C] pyruvate (Millipore Sigma, St. Louis, MO, USA), 10 μL of 15 mM trityl radical OX63 (Oxford Instruments, Abingdon, UK), and 0.4 μL of Gd3+ (Bracco Diagnostics, Monroe Township, NJ, USA) with an Oxford HyperSense instrument (Oxford Instruments, Abingdon, UK) at a temperature of 1.5 °K with microwave irradiation at 94 GHz. Once an average signal enhancement of 20,000-fold was achieved, a sample was prepared by dissolving the solid [1-^13^C] pyruvic acid in a rapidly heated 4 mL buffer comprised of 40 mM 2-amino-2-(hydroxymethyl)propane-1,3-diol (Millipore Sigma, St. Louis, MO, USA), 80 mM NaOH, 0.1 g/L ethylenediaminetetraacetic acid (Millipore Sigma, St. Louis, MO, USA), and 50 mM NaCl, pH 7.8. This solution had a final [1-^13^C] pyruvic acid concentration of 80 mM, and 200 μL of the solution was injected via the tail vein into mice in a magnetic resonance imaging scanner while they were anesthetized using 2% isoflurane.

All magnetic resonance imaging of mice was performed using a Bruker 7T horizontal bore preclinical MR scanner (Bruker BioSpin MRI GmbH, Ettingen, Germany) equipped with a medium-sized gradient coil. A 72-mm ^1^H volume coil (Bruker BioSpin MRI GmbH, Ettingen, Germany) was used to acquire anatomic images and perform accurate region-of-interest spectroscopy along with a ^13^C-urea phantom. Anatomical proton images of the mice were acquired using a rapid acquisition with relaxation enhancement T2-weighted pulse sequence. The proton images were acquired with a 57 ms echo time, 1444.38 ms repetition time, six signal averages, 9.5 ms echo spacing, 12 ms rapid acquisition with relaxation enhancement factor, 1.5 mm slice thickness, 256 × 192 image size, and 40 × 40 field of view. A ^13^C transmit/receive surface coil (ID: 35 mm; Doty Scientific Inc., Columbia, SC, USA) was employed to perform hyperpolarized ^13^C spectroscopy. All ^13^C spectra were acquired using a single-pulse, fast, low-angle shot sequence with a flip angle of 20°, image size of 2048 × 90, single slice with a thickness of 8–12 mm, and repetition time of 2 s over a period of 180 s to detect [1-^13^C] pyruvate and its metabolic product lactate. A ^13^C-urea phantom was used as a spectroscopic reference and tumor location reference [48]. All imaging procedures were performed in accordance with regulations of the MD Anderson Institutional Animal Care and Use Committee.

### 4.5. Mass Spectrometry

Quantification of fatty acids was performed using an Infinity II ultra-high performance LC instrument, a 6495 triple quadrupole mass spectrometer, and MassHunter quantitative analysis software (version 8.0.8.23.5; Agilent Technologies, Santa Clara, CA, USA). Murine gastric tumors stored at −80 °C were thawed, weighed, homogenized in methanol (Optima UHPLC-MS solvent; Thermo Fisher Scientific, Waltham, MA, USA) using a Bullet Blender tissue homogenizer (Next Advance, Troy, NY, USA), and centrifuged. The tumor supernatants were spiked with 1 ng/mL PA-5,6,7,8-^13^C_4_ (internal standard, cat. no. 605700; Sigma-Aldrich, St. Louis, MO, USA), dried using medical-grade nitrogen at 30 °C, and derivatized with 1-(3-amionopropyl)-3-bromoquinolinium bromide as described by Mochizuki et al. [49]. Derivatized fatty acids were separated using a Chromolith reversed-phase column (RP-18 endcapped 100–2 mm, cat. no. 152006; Sigma-Aldrich, St. Louis, MO, USA) with Optima UHPLC-MS mobile phase A (0.1% formic acid in water) and B (0.1% formic acid in methanol) gradients. The eluted fatty acids were introduced into an electrospray source in positive ion mode for analysis. The quantitative and qualitative m/z ion transitions for PA (m/z 503.1 to 296.3 and m/z 505.0 to 296.1), OA (m/z 529.0 to 322.3 and m/z 531.0 to 322.1), SA (m/z 533.1 to 324.2 and m/z 531.4 to 324.0), and LA (m/z 529.1 to 320.2 and m/z 527.1 to 320.4) were monitored using MassHunter quantitative analysis software (version 8.0.8.23.5; Agilent Technologies, Santa Clara, CA, USA).

### 4.6. Statistical Analysis

*p*-values were determined using multiple paired *t*-tests assuming that the Gaussian distribution and null hypothesis for the difference between paired values were consistent without correction for multiple comparisons using Prism software (version 9.0; GraphPad Software, La Jolla, CA, USA). For three-group comparisons, a two-way analysis of variance check was performed. Each cell mean of 10-week PPARD mice was compared with every other cell mean on 35-week PPARD mice and 55-week PPARD mice of that row (simple effects within rows) to calculate *p*-values in Prism. Principal component analysis score-plots were generated using BETA, large data edition (a web tool for visualizing clustering of multivariate data) (Figure 7) [50].

## Figures and Tables

**Figure 1 ijms-23-01645-f001:**
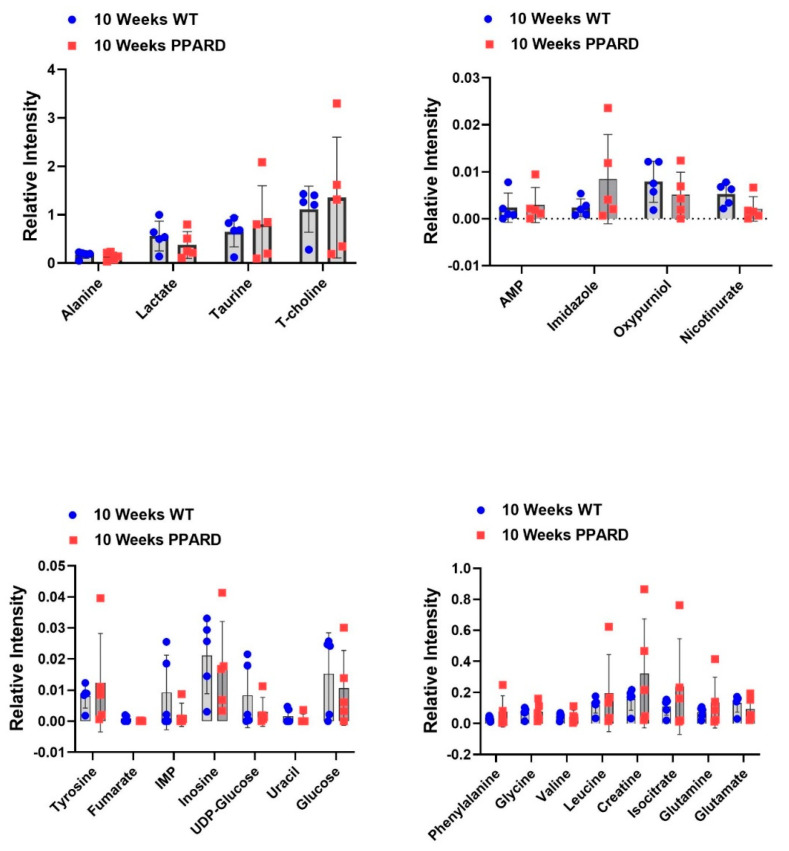
Differences in *ex vivo* metabolite concentrations in 10-week-old WT and PPARD mice plotted as interleaved scatter plots. No significant differences in metabolite levels between the two groups of mice were observed.

**Figure 2 ijms-23-01645-f002:**
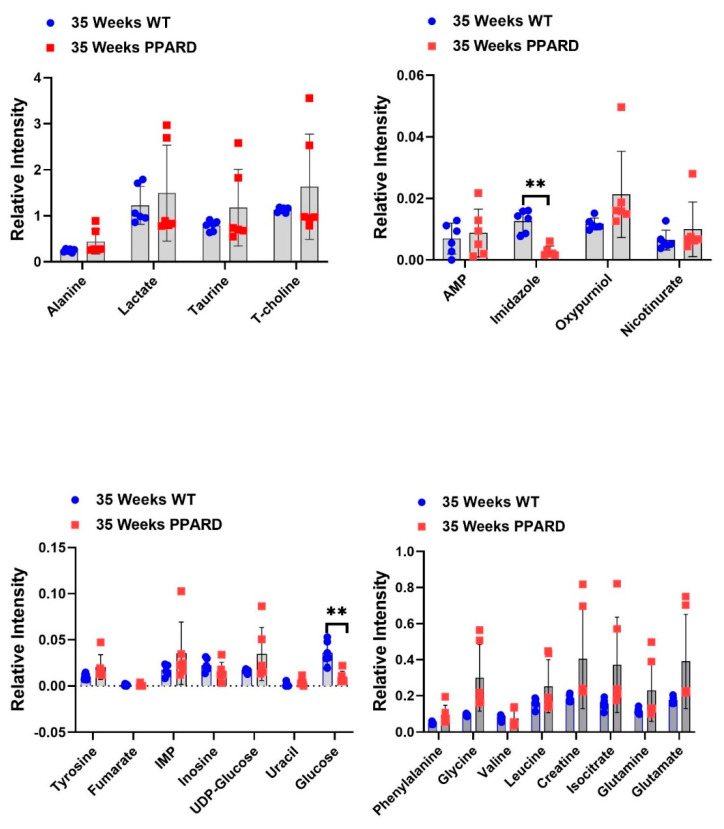
Differences in *ex vivo* metabolite concentrations in 35-week-old WT and PPARD mice plotted as interleaved scatter plots. The error bars represent SD. Significant differences in the glucose and imidazole levels were observed. ** *p* < 0.01.

**Figure 3 ijms-23-01645-f003:**
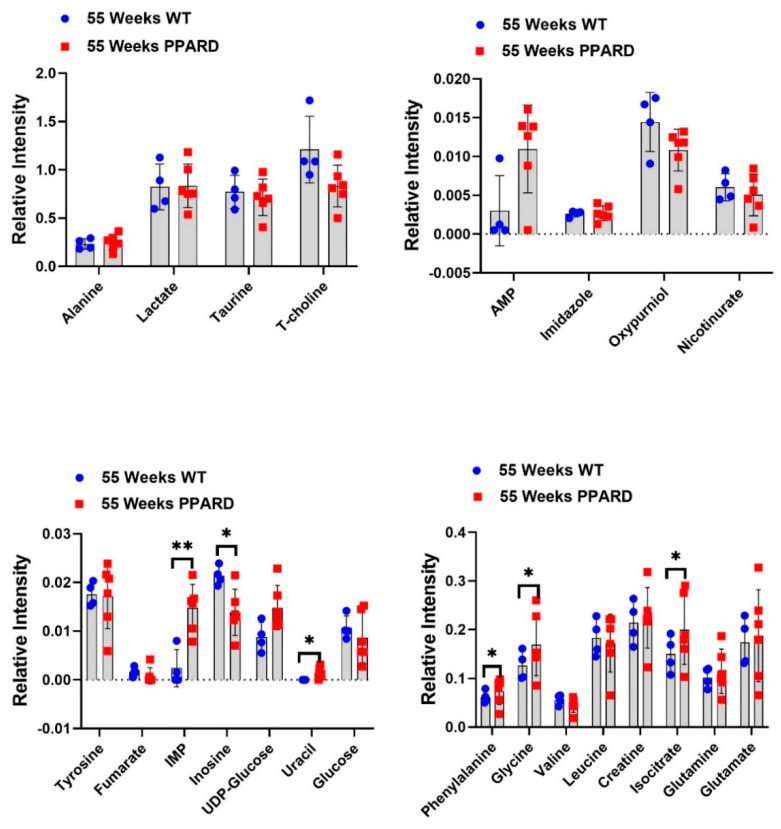
Differences in metabolite concentrations in 55-week-old PPARD and WT mice plotted as interleaved scatter plots. The error bars represent SD. Significant differences in IMP, inosine, uracil, phenylalanine, glycine, and isocitrate levels were observed. * *p* < 0.05; ** *p* < 0.01.

**Figure 4 ijms-23-01645-f004:**
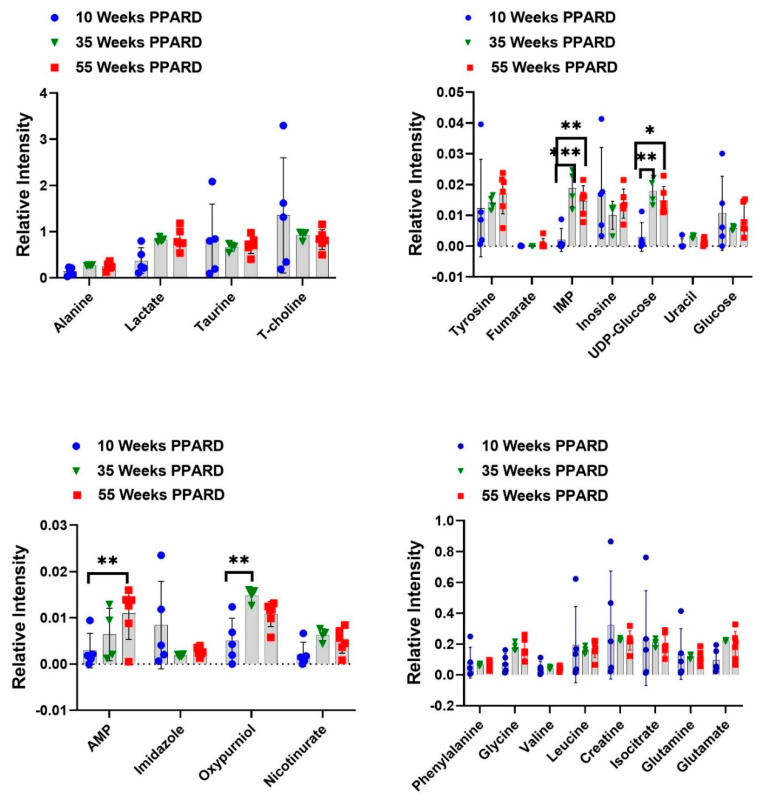
Differences in metabolite concentrations in PPARD mice during GC development at 10, 35, and 55 weeks of age plotted in interleaved scatter plots. The error bars represent SD. * *p* < 0.05; ** *p* < 0.01; *** *p* < 0.001.

**Figure 5 ijms-23-01645-f005:**
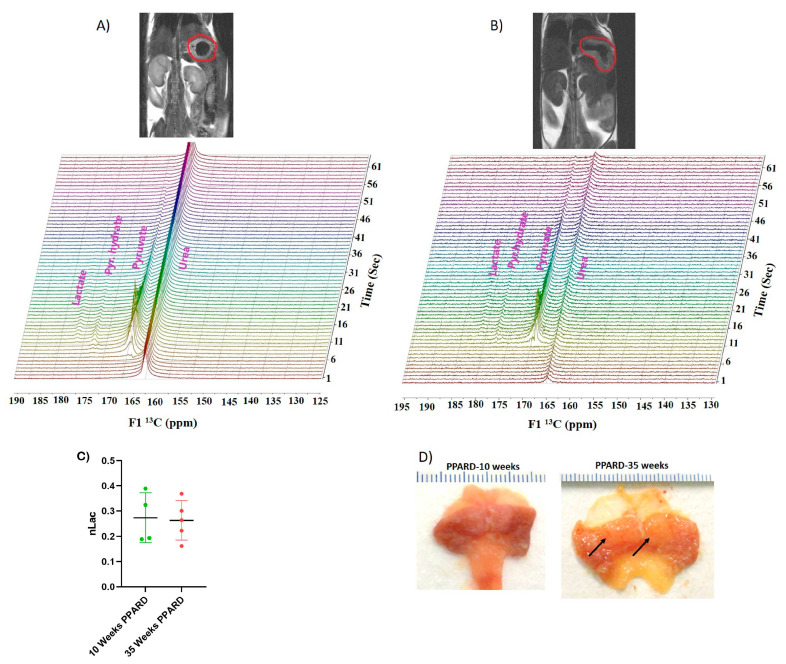
Real-time metabolic imaging with HP-MRI. Representative T2 proton MR images of 10-week-old (**A**) and 35-week-old (**B**) PPARD mice with their stomachs circled in red. Stacked HP-MR spectra are shown below the corresponding anatomical images. (**C**) Plot showing no significant changes in the nLac in PPARD mice from 10 to 35 weeks of age, indicating that glycolysis had a minimal role in proliferation of PPARD-driven GC. (**D**) PPARD mice experienced spontaneous development of large, invasive gastric tumors. Gross examination of these two stomachs removed from PPARD mice at 10 and 35 weeks of age showed tumor development. The arrows show tumor nodules within the corpus glandular mucosa of the stomach of the 35-week-old mouse.

**Figure 6 ijms-23-01645-f006:**
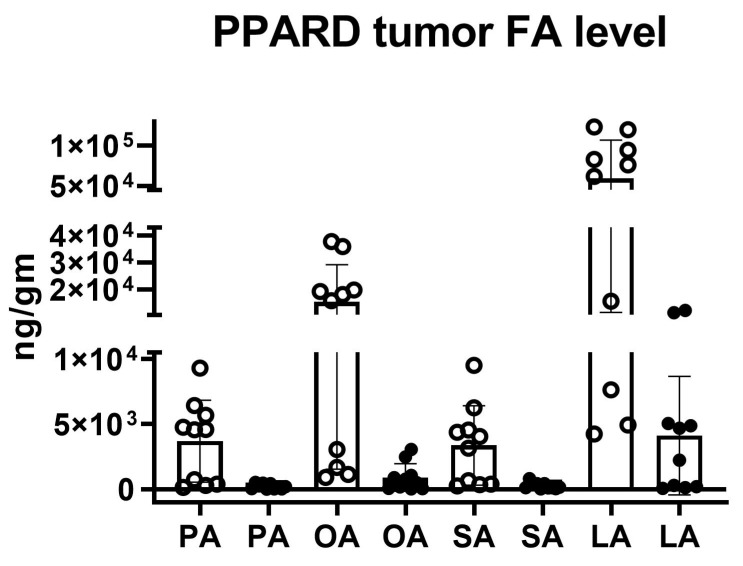
The fatty acid levels (in nanograms per gram) in gastric corpus tissues of PPARD mice at 10 weeks of age (before GC development) and 55 weeks of age (after advanced GC development). Palmitic acid (PA), oleic acid (OA), steric acid (SA), and linoleic acid (LA) levels in these tissues were lower in 55-week-old PPARD mice (closed circles) than in 10-week-old PPARD mice (open circles). Mann–Whitney *U* statistical analysis showed that the decreases in the levels of PA (*p* = 0.0029), OA (*p* = 0.0007), SA (*p* = 0.0028), and LA (*p* = 0.0015) were significant.

**Figure 7 ijms-23-01645-f007:**
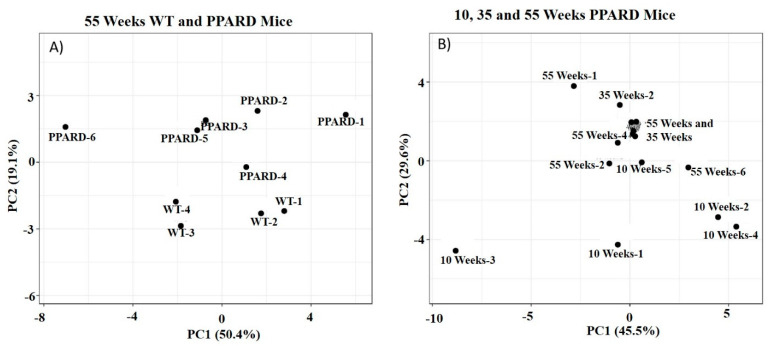
(**A**) Principal component analysis score-plot of 55-week-old WT and PPARD mouse gastric corpus tissue samples plotted as two orthogonal independent variables (PC1 and PC2), which constitute a total of 70%. In this plot, the WT and PPARD mice can be differentiated on the PC2 axis. (**B**) In this principal component analysis score-plot, 35- and 55-week-old PPARD mice overlap on both the PC1 and PC2 axes and cannot be differentiated, whereas 10-week-old PPARD mice can be differentiated from 35- and 55-week-old mice on the PC2 axis.

## Data Availability

The data that supports the findings are made available upon request to use appropriately to advance their research upon request to corresponding author.

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
