# Peer review of "Identifying the Metabolic Signatures of PPARD-Overexpressing Gastric Tumors"

_ijms, 2022, doi:10.3390/ijms23031645_

Round 1

Reviewer 1 Report

The manuscript is overall well written and scientifically sound. However, the authors skipped a section on "Materials and Methods" and therefore, the methods were not adequately described. I recommend the authors to include a section on the materials and methods in the revised version.

Author Response

The manuscript is overall well written and scientifically sound. However, the authors skipped a section on "Materials and Methods" and therefore, the methods were not adequately described. I recommend the authors to include a section on the materials and methods in the revised version.

Answer:  We thank reviewer for helping us to improve the quality of manuscript. 

The suggestion has been incorporated in the manuscript and additional information can be viewed in track changes and material methods subsections were highlighted in yellow.

Reviewer 2 Report

The presented paper considers the interesting study on cancer metabolism that show metabolic shift toward consumption of lipids instead of generally assumed glucose in the spontaneous mouse gastric cancer.

The data are novel and very well presented. The paper needs only minor editorial improvement:

  • Please verify the amount of LA described on p. 7: “… In stomach tissues of PPARD mice at age 10 weeks, the PPARD ligand fatty acid mean and (range) for PA and LA were 3,688.53 ng/mg (148.95 – 9,9292.75) and 3,688.53 ng/gm (4,245.91 – 123,336.7), respectively.
  • In Fig. 6 please include abbreviations for the lipid names, i.e. …. Palmitic (PA), oleic (OA), steri (SA), and linoleic (LA) acid….
  • Please check grammar, i.e
  • In discussion p. 8, the sentence: “… The PPARD found in skeletal muscle……”

Did you mean … The PPARD was found in skeletal muscle…?

  • In discussion p. 8, the sentence: …Time dependent changes in metabolism of PPARD mice model are also measured.”
  • In discussion p. 8, the sentence: “These data indicate…… of AMPK though…..”
  • In discussion p. 9, the sentence: “These results confirm that the important role …..
  • In Methods 4.1 p.9, the sentence: “In addition, PPARD mice….. were generated for dynamic……”

Did you mean “In addition, PPARD mice….. were subjected to dynamic……  ?

  • In Methods 4.2 p.9, the sentence: “The mixture was vortexed …. repeated for”
  • Results 4.5 p.10 – “water 0.1% formic acid” please change to “0.1% formic acid in water” and in further “methanol 0.1% formic acid” to “0.1% formic acid in methanol”.

Author Response

We thank reviewer for constructive comments and helping us to improve the quality of manuscript.

1) Please verify the amount of LA described on p. 7: “… In stomach tissues of PPARD mice at age 10 weeks, the PPARD ligand fatty acid mean and (range) for PA and LA were 3,688.53 ng/mg (148.95 – 9,9292.75) and 3,688.53 ng/gm (4,245.91 – 123,336.7), respectively.

Answer:  We have corrected this error and rewritten this paragraph in the Results section. 

In stomach tissues of 10-week-old PPARD mice, the mean PA and LA levels were 3,688.5 ng/g (range, 148.9-9,292.8 ng/g) and 18,510.0 ng/g (range, 4,245.9-123,336.7 ng/g), respectively. However, in stomach tissues of 55-week-old PPARD mice, the mean PA level decreased to 214.8 ng/g (range, 3.7-528.0 ng/g), whereas the mean LA level increased to 4,112.4 ng/g (range, 61.3-12,252.1 ng/g). Although in 55-week-old PPARD mice PA levels decreased 17-fold and LA levels decreased 4.5-fold, LA at the detected level of 4,112.4 ng/g may continue to bind and activate PPARD-driven transcription in the gastric tumors. The mean OA and SA levels in stomach tissues of 10-week-old PPARD mice were 15,355.1 ng/g (range, 904.4-37,808.0 ng/g) and 3,358.3 ng/g (range, 2,649.2-9,513.0 ng/g), respectively. The OA and SA levels in stomach tissues of 55-week-old PPARD mice were 16-fold and 13-fold lower, respectively, than those in stomach tissues of 10-week-old PPARD mice.

2) In Fig. 6 please include abbreviations for the lipid names, i.e. …. Palmitic (PA), oleic (OA), steri (SA), and linoleic (LA) acid….

Answer: Added

3) In discussion p. 8, the sentence: “… The PPARD found in skeletal muscle……”

Did you mean … The PPARD was found in skeletal muscle…?

Answer: The sentence has been replaced as- Also, researchers have studied the role of PPARD in obesity and diabetes and its action in adipose tissue, skeletal muscle, macrophages, and atherosclerosis [32].

4) In discussion p. 8, the sentence: …Time dependent changes in metabolism of PPARD mice model are also measured.”

Answer: The sentence has been revised as follows- We also measured time-dependent changes in metabolism in the PPARD mouse model using hyperpolarized [1-13C] pyruvate MR spectroscopy.

5) In discussion p. 8, the sentence: “These data indicate…… of AMPK though…..”

Answer:  This sentence has been revised as follows- These data demonstrated metabolic rewiring and may suggest an involvement of AMPK as it also stimulates β-oxidation of fatty acids[36-38]. One more reference (Ref 38) has added in support of our speculation - Cell metabolism 9 (2009): 407-16.

6) In discussion p. 9, the sentence: “These results confirm that the important role …..

Answer: The sentence has been modified as follows- Our NMR spectroscopy and LC-MS results confirmed the vital role of fatty acids and purine metabolism in GC development.

7) In Methods 4.1 p.9, the sentence: “In addition, PPARD mice….. were generated for dynamic……”

Answer: This sentence has been corrected as follows- PPARD mice at the ages of 10 and 35 weeks (n = 5 per group) were used for dynamic [1-13C] pyruvate metabolism measurements.

8) Did you mean “In addition, PPARD mice….. were subjected to dynamic……  ?

Answer:  We have revised the confusing sentence construction. This sentence has been revised as follows- PPARD mice at the ages of 10 and 35 weeks (n = 5 per group) were used for dynamic [1-13C] pyruvate metabolism measurements.

9) In Methods 4.2 p.9, the sentence: “The mixture was vortexed …. repeated for”

Answer: The sentence is revised as follows- The mixture was vortexed for 1 min and freeze-thawed; the vortexing and freezing-thawing were repeated three times.

10) Results 4.5 p.10 – “water 0.1% formic acid” please change to “0.1% formic acid in water” and in further “methanol 0.1% formic acid” to “0.1% formic acid in methanol”

Answer: Corrected